**Gridded surface O$_3$, NO$_x$, and CO abundances for model metrics from the South Korean ground station network**
Calum P. Wilson[1], Michael J. Prather[1]
[1]Department of Earth System Science, University of California (Irvine), Irvine, CA 92697, USA
*Correspondence to*: Calum P. Wilson (calumw@uci.edu)
**Abstract.** We present gridded surface air quality datasets over South Korea for three key species – ozone (O$_3$), carbon
monoxide (CO), and nitrogen oxides (NO$_x$) during the timeframe of the Korea–US Air Quality (KORUS–AQ) mission (May–
June 2016). The tenth–degree hourly averaged abundances are constructed from the 300+ air quality network sites using
inverse distance weighting with simple declustering. Cross–comparing the interpolated fields against the site data that was
used to create them reveals high prediction skill for O$_3$ (80%) throughout South Korea, and moderate skill (60%) for CO and
NO$_x$ on average in densely observed regions after individual mean bias corrections. The gridded O$_3$ and CO interpolations
predict the NASA DC–8 observations in the planetary boundary layer (PBL) with high skill (80%) in the Seoul Metropolitan
Area (SMA) after subtracting the mean bias. DC–8 NO$_x$ observations were much less predictable on account of consistently
negative vertical gradients within the PBL. Our gridded products capture the mean and variability of O$_3$ throughout South
Korea, and of CO and surface NO$_x$ in most site–dense urban centres (SMA, Cheongju, Gwangju, Daegu, Changwon, and
Busan).
**1 Introduction**
Air quality control has become a priority in the Republic of Korea following an upward trend in ozone (O$_3$) pollution in all
major cities since the 1980s (Susaya et al., 2013). In May–June 2016, the Korea–US Air Quality (KORUS–AQ) mission was
launched with the goal of improving knowledge of the factors controlling Korean air pollution; this mission gathered extensive
observational data via aircraft, ground stations, ships, and remote sensing (Crawford et al., 2021).
Comparisons of modelled grid–cell values (i.e., averages) with point data from station sites remains awkward,
especially in high–emission environments with high sub–grid and temporal variability. Ground site comparisons in South
Korea have thus far used the arithmetic mean of sites within a grid cell or ungridded quantile analysis (Lennartson et al., 2018;
Peterson et al., 2019; Eck et al., 2020; Jordan et al., 2020; Schroeder et al., 2020; Park et al., 2021; Oak et al., 2022; Travis et
al., 2022), but these unweighted means can be biased by site clustering, and they lose information outside the cells. In this
work we develop a gridded dataset of key surface–level pollutants (in this case, O$_3$, NO$_x$, CO) observed during the KORUS–
AQ timeframe. In contrast to arithmetic means, we apply Inverse Distance Weighting (IDW) interpolations (Shepard, 1968)
improved by Schnell et al. (2014) to create a country–wide continuous mapping of the National Institute of Environmental
Research (NIER) ground site data. We subsequently integrate the interpolated field over a 0.1°x0.1° grid. To evaluate the

interpolation, we predict NIER station measurements using the leave–one–out cross validation method; we predict observations from two research sites (Olympic Park and Taehwa Forest) to verify instrumental cohesion; and, we compare our gridded fields with DC–8 observations within the planetary boundary layer (PBL) to gauge how well the data products reproduce upper PBL abundances. In addition to providing gridded PBL datasets, we discuss the applicability and limitations of our methodology for each key species.

The observational data sets are described in Section 2, and the methods in Section 3. Results are summarized in Section 4. Conclusions and recommendations are presented in Section 5.

## 2 KORUS–AQ data

All KORUS–AQ datasets introduced in this section including the raw five–minute NIER station data are available via https://doi.org/10.5067/Suborbital/KORUSAQ/DATA01. The NIER station data is access–controlled, but we record our processed hourly–averaged NIER station data and our quality control flags for the raw data in our repository (see Wilson, 2024).

## 2.1 NIER air quality stations

The AirKorea monitoring network (https://www.airkorea.or.kr/eng) provided ground measurements of the key species averaged every 5 minutes at 323 stations across South Korea, of which 319 reported $O_3$, 311 reported CO, and 321 reported $NO_x$ (Fig. 1). We calculate hourly median readings centred on the hour for each station, but omit clearly erroneous $O_3$ and $NO_x$ dropouts from the average. High outliers exceeding five standard deviations above the mean of a weekly period were also discarded along with dropouts, which were manifest as stably low concentrations (1–4 ppb) persisting for multiple hours in stark contrast with the typical variability at the site. We were able to flag most dropouts algorithmically by analyzing the cumulative density functions (CDFs) of the station data partitioned into non–overlapping weekly intervals; improbably frequent low data often featured flat empirical gradients (less than $100^{th}$ of the median CDF gradient) at the tail of the CDF. This technique proved insufficient at some stations however, and so we manually removed dropouts that were not flagged by our algorithm, as did Eck et al, 2020. The NIER instruments and procedures are not well documented and there remain some oddities: CO was reported with 1 ppb precision at 68 sites, and with 100 ppb precision at the remaining 250 sites.

## 2.2 Research stations

### 2.2.1 Olympic Park

The Olympic Park research station lies at the southeast edge of Seoul at 37.5216°N, 127.1242°E, 30 m above sea level, and served as a reference for ground–level Seoul pollution during the KORUS–AQ campaign (red star in Fig. 1). Hourly averages for the key species were recorded using $NO_x$–Ecotech EC9841, CO–Ecotech EC9830, and $O_3$–Ecotech EC9810 instruments (PI: Cho Seogu) during the KORUS–AQ period (10 May 01:00:00 to 18 June 00:00:00 LT). As Olympic Park station has four

proximal NIER stations within 5 km, reproducing this research station data from the NIER interpolation should be a test of the
small scale variability of Seoul pollution provided the instruments are well calibrated.

**2.2.2 Taehwa Forest**

The Taehwa Forest wilderness site lies 30 km southeast of Olympic Park at 37.3123°N, 127.3105°E and at 200 m elevation
(blue star in Fig. 1). It was used primarily to investigate the mixing of urban Seoul pollution with the biogenic volatile organic
compounds (BVOCs) of the forest. The three key species were measured by the existing NIER instruments (PI: Youngjae Li),
but supplemented by a Thermo Scientific 42i instrument for NO and a Cavity Ring–Down Spectroscopy for $NO_2$ (PI: Kim
Saewung, Kim et al., 2022).

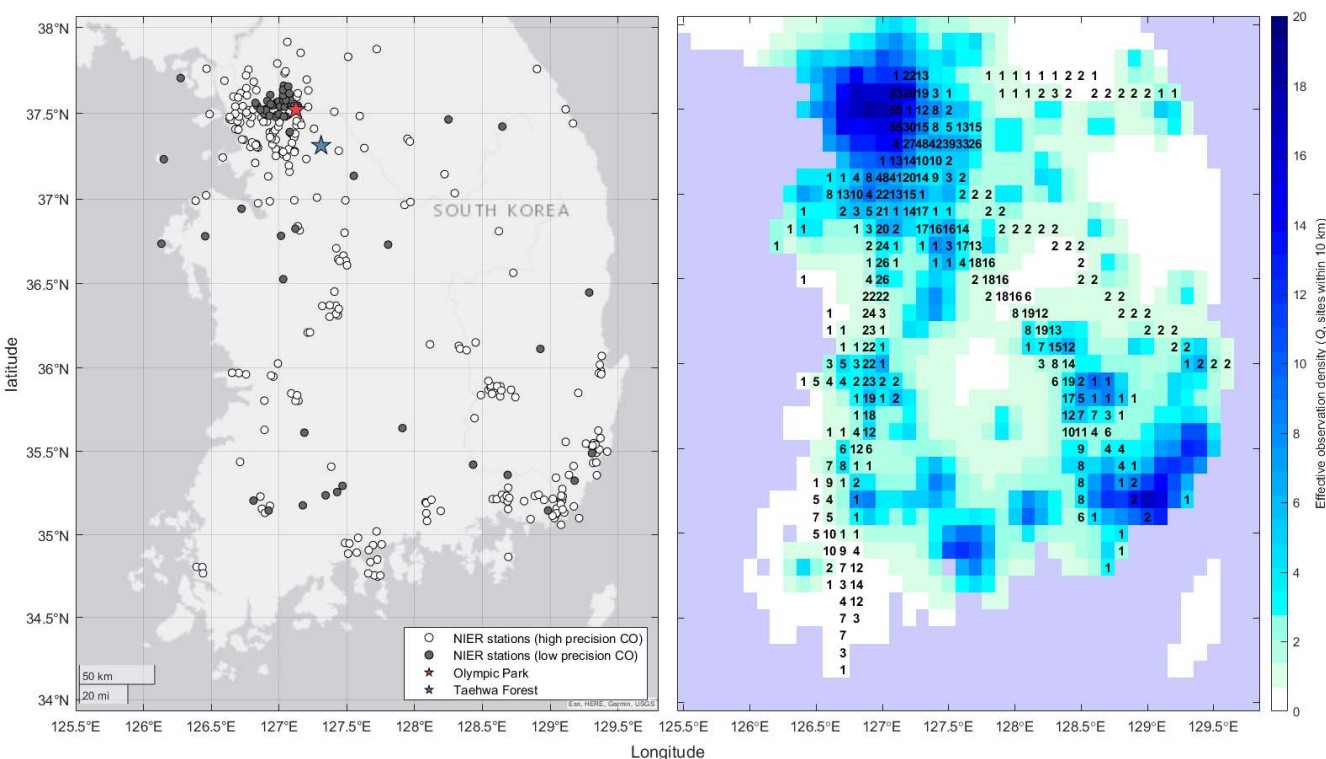

**Figure 1:** (**Left**) The geographical distribution of NIER ground stations and the two surface research stations operating during the KORUS–AQ campaign. High–precision stations (white circles) recorded CO at 1 ppb precision; low–precision stations (grey circles) recorded CO at 100 ppb increments. (**Right**) *Effective NIER station density (colour) within a 10 km radius* ($Q$, see Eq. (3)) gridded over 0.1°x0.1° cells. The number of contiguous DC–8 flight transects through each box in the PBL is printed in each cell. The aircraft radar altitude was evaluated against the ERA5 PBL height (based on hourly 0.25°x0.25° gridded data, Hersbach et al., 2023). The ERA5 data was interpolated in time to match the aircraft data.

**2.3 NASA DC–8**

The DC–8 aircraft routinely profiled the air over Taehwa Forest via loop manoeuvres in the morning and afternoon on flight
days between 2 May 2016 and 11 June 2016. It sampled other regions above South Korea and the Yellow Sea according to
pollution plume transport and cloud forecasts. We use the 10 s merged data our three key species: $O_3$, $NO$, and $NO_2$ were
measured with a 4–channel chemiluminescence instrument (Weinheimer et al., 1994); and $CO$, by Differential Absorption
Carbon monOxide Measurement (DACOM) (Sachse et al., 1991). We also use the 10 s data for latitude, longitude, radar
altitude, UTC time, and potential temperature (PI: Melissa Yang). From the DC–8 potential temperature measurements and
ERA5 surface data (Fig. A1) we can show that the ERA5 PBL heights accurately select DC–8 observations that are
adiabatically mixed from the surface (i.e., $d\theta/dz \sim 0$), which is confirmed by the afternoon $O_3$ and $CO$ profiles (Fig. A2). To
determine when the aircraft was in the PBL and thus could be compared with the interpolated surface map, we use the ERA5
PBL height data from reanalysis (hourly, 0.25°x0.25° grid, Hersbach et al., 2023). This approach is more accurate than simply
assuming that all DC–8 observations below 1.5 km radar altitude fall within the PBL (e.g., Oak et al., 2019).
**3 Methods**
Interpolation techniques compute an objective estimate $Z'(x, t)$ of a field $Z(x, t)$ at any geographic location $x$ and time $t$ as a
weighted mean of observations $Z_k(t)$ at stations indexed by $k$ with weights $w_k(x)$:
$$Z'(x, t) = \sum_k [w_k(x) Z_k(t)] / \sum_k w_k(x) \tag{1}$$
Ordinary Kriging and Inverse Distance Weighting (IDW) are two common interpolation methods that operate by this premise
but differ in how the station weights ($w_k$) are calculated (Matheron, 1963; Shepard, 1968). Kriging is a family of statistical
techniques based on the supposition that phenomena are autocorrelated in space, relying on an empirical distance–based
covariance model of $Z(x, t)$ determined from the station data. In our work we find minimal Pearson's correlation between
pairwise site covariance and proximity for any of the key species, so we opt for the modified IDW approach of Schnell et al.

96    (2014).

We examine the autocovariance functions of the sites to characterize the temporal variability of each key species at
five–minute resolution. The autocovariance function for a given phase shift (*e.g.* ten minutes) is defined as the covariance of
a time series of data with itself, but phase–shifted by ten minutes. The $O_3$ site autocovariance functions feature strong diurnal
cycles, preserving on average half of the respective site variances for a twenty–four–hour phase shift, compared with 35% for
$CO$ and $NO_x$. Sub–ten–minute variability accounts for 3% of the $O_3$ site variance and 12% of the $CO$ and $NO_x$ variance on
average. Heterogeneous emissions and micrometeorology may be attributed to some of the $CO$ and $NO_x$ short timescale
variability seen in some Seoul sites and many $CO$–measuring sites in Gwangyang–Yeosu–Suncheon zone, while rural sites
suggest a possible instrumental noise component to the $NO_x$ variability (See Fig. B of Appendix B).
**3.1 Inverse Distance Weighting**
In IDW techniques, weights are calculated from the reciprocal distances between estimation point $x$ and the station coordinates
$x_k$, scaled by the exponent $\beta$. The greater density of observations in some regions creates a source of oversampling bias.
Schnell et al. (2014) address this clustering effect by reducing all station weights by $M_k$, the number of other stations within
distance $D$ of site $k$, discounting sites with missing readings on an hourly basis. In order to smooth the spatial heterogeneity
in $Z'(x,t)$ at small length scales, the distance $D$ also serves as the minimum cutoff of $x-x_k$, and hence determines the
maximum weighting $w_k(x)$ of any nearby station. $L$ is a maximum cutoff of $x-x_k$ used to reduce excess calculations for
extremely distant and unimportant sites. The weight formulae are summarized in Eq. (2):
$$w_k(x) = \frac{D^{-\beta}}{M_k} \qquad\qquad x - x_k \leq D$$

$$w_k(x) = \frac{(x-x_k)^{-\beta}}{M_k} \qquad\qquad D < x - x_k \leq L \qquad\qquad (2)$$

$$w_k(x) = 0 \qquad\qquad x - x_k > L$$

Our NIER station data consists of $k \in \{1, 2, \ldots, 323\}$ locations (Fig. 1) and $t \in \{1, 2, \ldots, 936\}$ hourly observations (10 May
01:00:00 to 18 June 00:00:00 LT) for each of our three key species ($O_3$, CO, $NO_x$) with some unreported or erroneous data.
We optimize $\beta$ and $D$ for each key species $Z$ by randomly removing a fifth of the stations from the algorithm and then predicting
the abundance at each missing station $k'$, adjusting each parameter until a 2D minimum was reached. In minimizing the total
root–mean–square error between predictions $Z'_{k'}(t)$ and observations $Z_{k'}(t)$ over the time series, we find similar optimal
values for each species ($\beta \sim 2$, $D \sim 5$ km, $L \sim 80$ km), with no significant improvement for larger $L$. The *effective density of*
*observations* $Q(x)$ is defined as the effective number of NIER sites within a 10 km radius of $x$ in Eq. (3) (also called *Quality*
*of prediction*, Eq. (5) of Schnell et al., 2014). We expect $Q$ to correlate with prediction accuracy:
$$Q(x) = 10^{\beta} \sum_k w_k(x) \qquad\qquad (3)$$

Compared with the arithmetic mean gridding, the gridded IDW observations show no significant mean bias. However,
significant mean absolute deviations can be seen in the densely observed ($Q > 10$) Seoul Metropolitan Area and Southeast
coastal cities. In such regions, the mean absolute deviations for $O_3$, CO, and $NO_x$ are 5 ppb, 58 ppb, and 9 ppb respectively.
Conversely, the most sparsely measured regions show the best agreement between the two methods due to mutual sampling
of a single station. Both methods are fundamentally limited by sampling density, particularly in urban centres with high
spatial emission variability. The differences seen in the most densely observed regions highlight the instability of the
arithmetic mean method with respect to grid size and boundary manipulation.
**3.2 Statistical techniques**
To evaluate the accuracy and predictive capability of an interpolation, we examine the error $E(t)$ in a time series of predictions
$Pre(t)$ and observations $Obs(t)$ at a given location for a given species with all time points equally weighted equally. We
calculate a sequence of three error series defined as follows:
$$E1(t) = Pre(t) - Obs(t)$$

$$E2(t) = Pre(t) - Obs(t) - (\overline{Pre(t)} - \overline{Obs(t)}) \qquad\qquad (4)$$

$$E3(t) = b\,Pre(t) - Obs(t) - (b\,\overline{Pre(t)} - \overline{Obs(t)})$$

Where $E1(t)$ is the absolute error in the predictions, $E2(t)$ is the error after correcting for the mean prediction bias $(\overline{Pre(t)} -$
$\overline{Obs(t)})$ and $E3(t)$ is the error relative to a simple linear regression (LR) model of $Pre(t)$ vs. $Obs(t)$ fitted by ordinary least
squares, i.e., after correcting for mean bias and slope ($b$). We then apply the *coefficient of determination* to compute the fraction
of the observed sample variance, $Var(Obs(t))$, explained by e.g. the raw predictions ($E1(t)$):

$$R^2_{E1} = 1 - \frac{Mean(E1(t)^2)}{Var(Obs(t))} \qquad (5)$$

And do similarly for $E2(t)$ and $E3(t)$. $R^2_{E1}$ is a *predictive accuracy* statistic that ranges from minus infinity to one and is
identical to the forecast skill score referenced to the mean of observations (Murphy, 1988). $R^2_{E2}$ describes how well the
predictions capture the temporal variability in the observations regardless of any mean bias and has the same range as $R^2_{E1}$.
$R^2_{E3}$ is the common definition of $R^2$ in regression analysis and ranges from zero to one due to the fitting constraint. $R^2_{E3}$
describes the *predictability* of the observations from the LR model regardless of any difference in the mean or variance of
$Pre(t)$ and $Obs(t)$. A score of zero for a given $R^2_E$ is equivalent to predicting a static mean of observations across the time
domain. The maximum score for $R^2_{E1}$ and $R^2_{E2}$ is limited by the interpolation variance, which is typically damped relative to
the contributing stations, especially in regions with highly heterogeneous emissions. Figure 2 (right–hand side) suggests the
average station predictability ($R^2_{E1}$ and $R^2_{E2}$) score has an upper bound of around 0.9 for $O_3$ and 0.8 for CO and $NO_x$.
**3.2.1 Leave–one–out cross validation.**
In this trial, we sequentially remove each station $k$, then interpolate (predict) its value from the remaining stations: $Pre(k,t) =$
$Z'(k,t)$, where $Obs(k,t) = Z(k,t)$ (see Eq. (4); Brauer et al, 2003; Hochadel et al., 2006). A perfect interpolation would
accurately reproduce the mean and standard deviation of the measurements, indicating (1) no mean bias error and (2)
preservation of daily maximae and minimae. Our optimized IDW interpolation has clearly worked well in terms of mean bias
(left half of Fig. 2). The box quartiles and non–outlier whiskers (i.e., the full range of values within one–and–a–half
interquartile ranges from the outer quartiles) are well centred on zero bias, with the spread broadening from $O_3$ to CO to $NO_x$.
The symmetry of the whiskers comes from the case where two sites, distant from the remaining sites but near one another, are
the only sites used to interpolate one another and hence if one site has twice the mean value of another, we get symmetric plus–
minus biases for each site. The median of the mean $NO_x$ site biases is +13%, and this appears to be an artefact of low $NO_x$
abundances in rural ($Q < 5$) locations. The absolute mean $NO_x$ bias averages –0.6 ppb (urban –3.0 ppb, rural +6.5 ppb).
Incoherence among nearby urban stations combines to dampen the interpolation variability, especially for CO and $NO_x$, which
feature independent high spatial variability from local sources. This is shown on the right half of Fig. 2, where most of the
standard deviation ratio quantiles lie below unity. We believe this reduced standard deviation in the prediction time series
better represents the average over a grid cell that contains several incoherent sites.

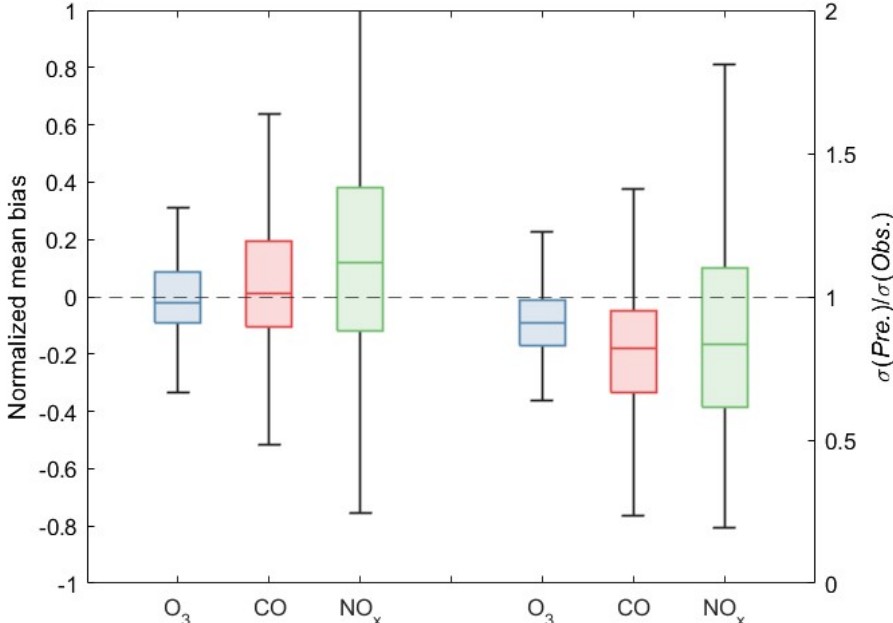


**Figure 2: (left)** Box plots of normalized mean bias: $NMB(k) = Mean(Pre(k, t) - Obs(k, t))/Mean(Obs(k, t))$ and **(right)** standard deviation ratio $\sigma(Pre(k, t))/\sigma(Obs(k, t))$ for interpolated time series at each NIER site using leave–one–out cross validation. Whiskers show the range of non–outliers, where outliers are data beyond one–and–a–half interquartile ranges from the outer quartiles. Results are shown for $O_3$ (blue), CO (red), and $NO_x$ (green). Mean bias is normalized by the observed mean, and the ratio of standard deviations is analogous to the gradient of a linear regression.

174

The sequence of $R^2_E$ scores ($E1$–3) for each site and each species are shown in Fig. 3. The $O_3$ scores (top row) are consistently

high across the sequence. $R^2_{E1}$ through $R^2_{E3}$ scores for $O_3$ indicate that the $O_3$ interpolation was accurate and unbiased at almost

all NIER stations in South Korea. For CO (middle row) and $NO_x$ (bottom row), there is an improvement in absolute prediction

accuracy ($R^2_{E1}$) as the density of observations ($Q$) increases, and further improvement after correcting the mean bias in the

predictions ($R^2_{E2}$). The linear regression models ($R^2_{E3}$) offer an obvious improvement to predictability in rural regions (low $Q$)

where information is lacking, but no significant improvement in well sampled urban regions (high $Q$). With no large net mean

bias for any key species (Fig. 2), we assert that the average of our interpolations should capture the mean and possibly the

variability of a well–mixed gridded domain. We test this assertion later using aircraft PBL observations averaged into 0.1°x0.1°

cells. The high range of $R^2_E$ values for $NO_x$ and CO, even where $Q > 10$, suggests that absolute mean error in the prediction is

a problem for many sites, implying they are driven by very small scale (<1 km) local emissions in contrast to $O_3$, which is not

emitted directly. For $NO_x$, the sequence from $E1$ to $E2$ greatly improves the prediction accuracy. For CO, there remains a large

fraction of unpredictable sites, often with very high standard deviations (dark red circles), implying large nearby emissions.

Figure 4 (top–middle and top–right panels) shows the clustering of such sites for CO and $NO_x$ in Daejeon (central–western

South Korea) and in the southern coastal cities of Gwangyang, Yeosu, Suncheon, Jinju, and Ulsan (no $NO_x$ data). When we

compare the LOOCV performance of $NO_x$ with the complete interpolation (*i.e.*, no stations omitted, see bottom panel of Fig.
4), we see $R^2_{E2}$ scores change by $> 0.4$ at *e.g.* rural sites, the Busan shoreline, and the manufacturing district sites of Northern
Yeosu and Eastern Gwangyang. These discrepancies indicate under–sampling of $NO_x$ across rural South Korea and in some
urban districts with locally contrasting emission activity.

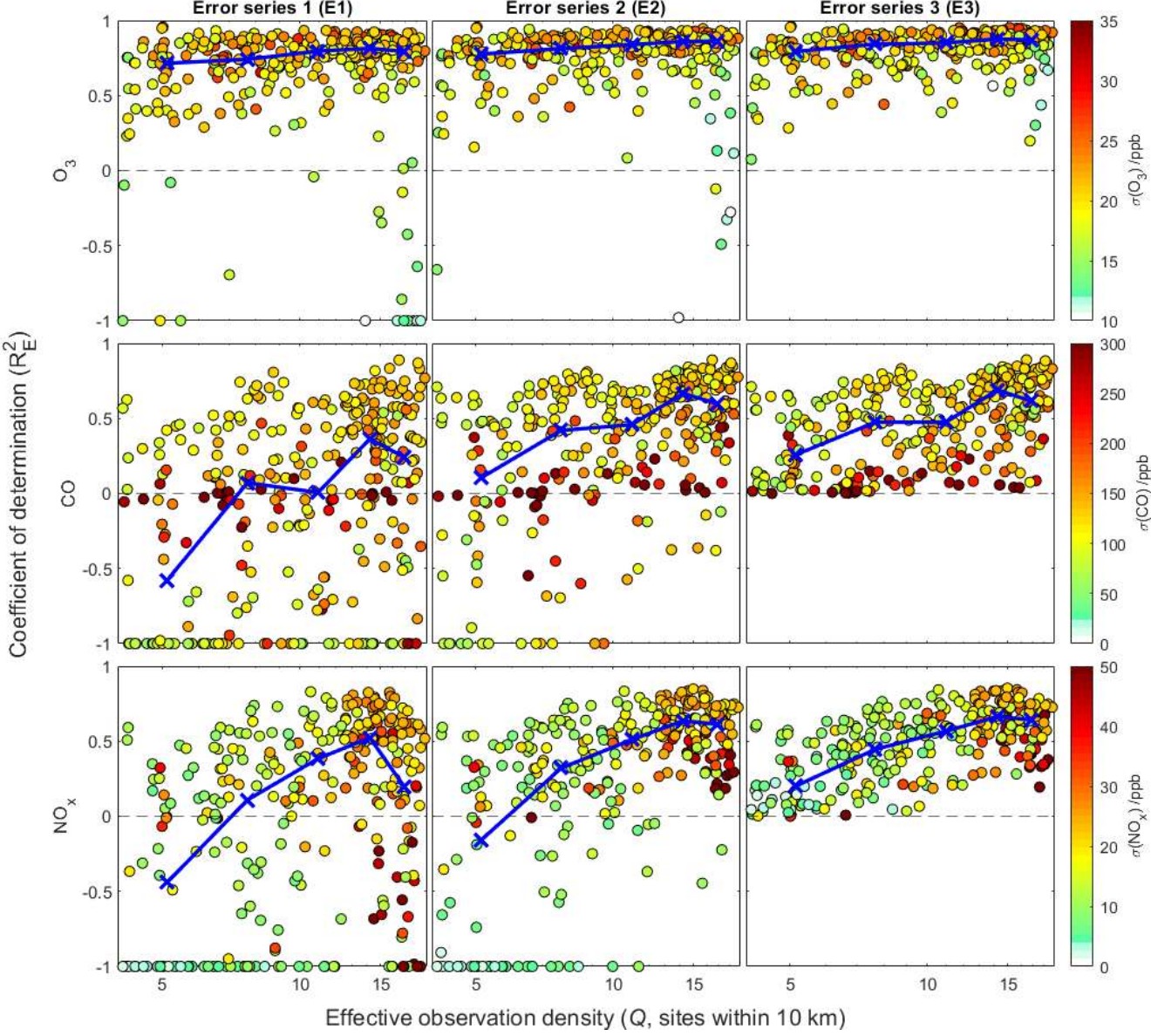


**Figure 3:** Generalized coefficient of determinations ($R^2_E$, Eq. (5)) for NIER station predictions vs. the effective density of nearby observations ($Q$, effective number of sites in a 10 km radius). The three columns show the sequence $R^2_{E1}$, $R^2_{E2}$, and $R^2_{E3}$. The three rows are for the species $O_3$ (**top**), CO (**middle**), and $NO_x$ (**bottom**). The calculations use the leave–one–out cross validation at each NIER station (circles) coloured by the standard deviation of observations. The blue conjoined crosses show the median $R^2_E$ values for five percentile partitions of $Q$: 0–20%, 20–40%, 40–60%, 60–80%, and 80–100%.

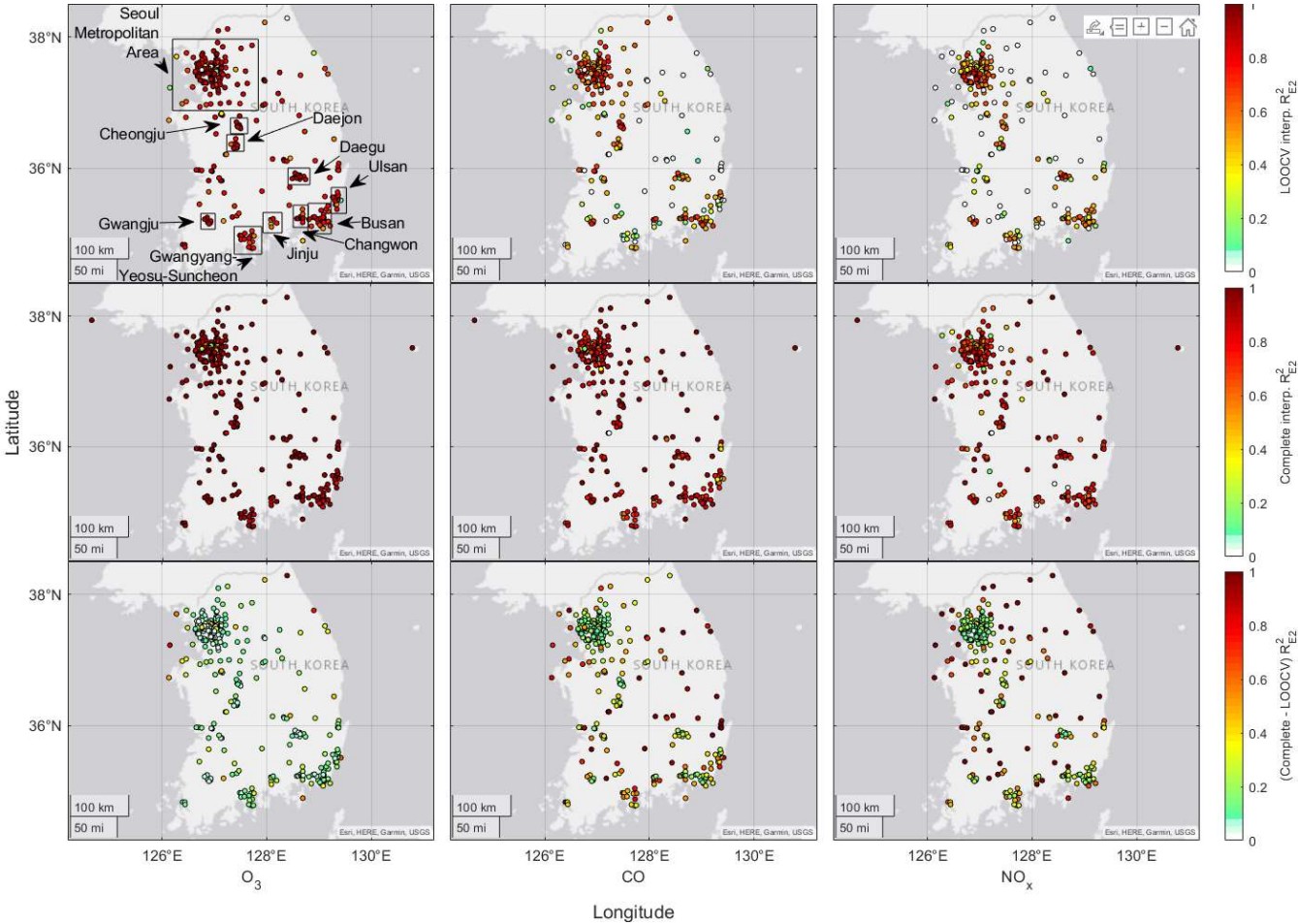

**Figure 4:** The geographical distribution of NIER station prediction accuracies with the mean prediction bias removed from each station ($R^2_{E2}$, Eqs. (4) and (5)), shown for the three key species: $O_3$ (**left**), CO (**middle**), and $NO_x$ (**right**). **Top:** $R^2_{E2}$ scores for the LOOCV interpolations. **Middle:** $R^2_{E2}$ scores for the complete interpolation, *i.e.* without omitting the data from predicted sites. **Bottom:** The difference between the complete– and LOOCV– interpolation $R^2_{E2}$ scores. Negative $R^2_{E2}$ values are truncated to zero. Approximate city bounds are shown via text and boxes in the top–left panel.

We have additionally compared the interpolation accuracy during the four meteorological phases presented by Peterson et al. (2019), *i.e.*, dynamic, anticyclone, low–level transport, and rex blocking, although we did not identify any obvious patterns across the phases (see Fig. C1 of Appendix C). An alternative test of meteorological influence is daytime (07:00 to 20:00 LT) *vs.* nighttime (21:00 to 06:00 LT) predictability, *i.e.*, prevailingly turbulent *vs.* stably stratified atmospheric surface conditions. Ozone and $NO_x$ showed greater LOOCV predictability ($R^2_{E2}$) during daytime than nighttime by around +0.1 and +0.05 respectively, but not CO (see Fig. C2). More efficient turbulent mixing during daytime vs. nighttime likely smoothed some of the small–scale emission heterogeneity, resulting in more predictable fields.

### 3.3 Gridded air quality data

A major objective of this study was to obtain grid–cell averages (0.1°x0.1°, approx. 10 km x 10 km) for testing regional air quality models. Within each 0.1°x0.1° cell, we interpolate the key species to twenty–five points on a 0.02°x0.02° grid centred in the cell, and then average these values. The averages do not account for latitudinal differences in quadrangle areas, which are minor for South Korean latitudes. We apply the same treatment to the density of observations to produce the gridded $Q$ values as seen in Fig. 1 (right–hand side).

### 3.4 Aircraft cell averages

We collect the measurements of $O_3$, CO, and $NO_x$ from NASA DC–8 taken over land at radar altitudes below the PBL heights taken from the ERA5 data. The DC–8 measurements used here are 10 second merges corresponding to approximately 1 km flight segments $S \in \{1, 2, \ldots, 13942\}$. To compare the segments with the gridded site data, we average the contiguous segments through each grid cell to produce transect–averaged observations $Obs(T)$, where transects $T \in \{1, 2, \ldots, 2106\}$ contain around seven segments whose midpoints lie in the cell bounds. For the prediction set $Pre(T)$, we interpolate the traversed cells in time to match the mean aircraft time of flight during the respective transects. The number of transects through each cell is indicated by the gridded numbers in Fig. 1 (right–hand side).

### 4 Results

**Table 1:** The generalized coefficients of determination $R^2_{E1}$, $R^2_{E2}$, and $R^2_{E3}$ (Eq. (5)) for predictions vs. measurements at research stations (Olympic Park and Taehwa Forest) and along flight transects in the PBL. Each flight transect is a median of contiguous 10 s observations through a grid cell (See Fig. 1 for sampling distribution and Fig. 5 for scatter plots), and the predictions are gridded values interpolated linearly in time to match the aircraft time of flight, then averaged. $E1$, $E2$, and $E3$ are time series of prediction errors defined in Eq. (4). $NO_x$ measurements at Taehwa Forest are taken from Kim et al., 2022.

| Species | Olympic Park | | | Taehwa Forest | | | DC–8 (all transects) | | | DC–8 ($Q > 10$ transects) | | |
|---|---|---|---|---|---|---|---|---|---|---|---|---|
| | $R^2_{E1}$ | $R^2_{E2}$ | $R^2_{E3}$ | $R^2_{E1}$ | $R^2_{E2}$ | $R^2_{E3}$ | $R^2_{E1}$ | $R^2_{E2}$ | $R^2_{E3}$ | $R^2_{E1}$ | $R^2_{E2}$ | $R^2_{E3}$ |
| $O_3$ | 0.90 | 0.92 | 0.96 | 0.68 | 0.82 | 0.82 | 0.02 | 0.69 | 0.69 | 0.26 | 0.81 | 0.90 |
| CO | 0.73 | 0.75 | 0.76 | –2.70 | 0.69 | 0.71 | –2.20 | 0.28 | 0.41 | –0.91 | 0.83 | 0.84 |
| $NO_x$ | 0.67 | 0.68 | 0.68 | –12.0 | –3.60 | 0.00 | –2.60 | 0.34 | 0.62 | –0.84 | 0.51 | 0.73 |

### 4.1 Research site prediction

Research stations provide case studies where the quality of measurements is carefully controlled, and so instrumental drift, noise, and biases are minimized. For each key species, we compare the NIER station data interpolated to the coordinates of the research stations, either at Olympic Park or Taehwa Forest, against the research station instruments (Fig. 5). Olympic Park and Taehwa Forest have effective sampling densities ($Q$) of 16 and 6 stations per 10 km respectively. Figure 5 shows accurate

prediction of $O_3$ at both sites with predictably more scatter at Taehwa Forest where less information was available. We see a
similar pattern for CO, but with a mean bias (predicted NIER interpolated value minus research instrument measurement) of
+100 ppb at Taehwa Forest. $NO_x$ is predicted reasonably well at Olympic Park except in the highest measured range (>100
ppb), but predictions appear random at Taehwa Forest. Table 1 indicates excellent prediction accuracy at Olympic Park for all
species ($R^2_{E1}$), and at Taehwa Forest for $O_3$. At Taehwa Forest, CO prediction improves when mean biases are removed ($R^2_{E2}$),
but $NO_x$ remains unpredictable. The linear regressions ($R^2_{E3}$) lead to very little improvement over mean bias correction ($R^2_{E2}$),
implying that the temporal variability measured by the research stations was well captured. High $R^2_{E1}$ scores suggest good co–
calibration between the Olympic Park instruments and surrounding NIER instruments. We are unable to characterize the mean
biases at Taehwa Forest.
As an isolated wilderness site, Taehwa Forest presents a unique problem for interpolating $NO_x$ values based on NIER
stations. The closest three NIER sites surround the forest station at a distance of around 10–15 km, and all are subject to NOx
roadside emissions, thus our interpolation maps these high–NOx values into the relatively NOx–depleted forest.

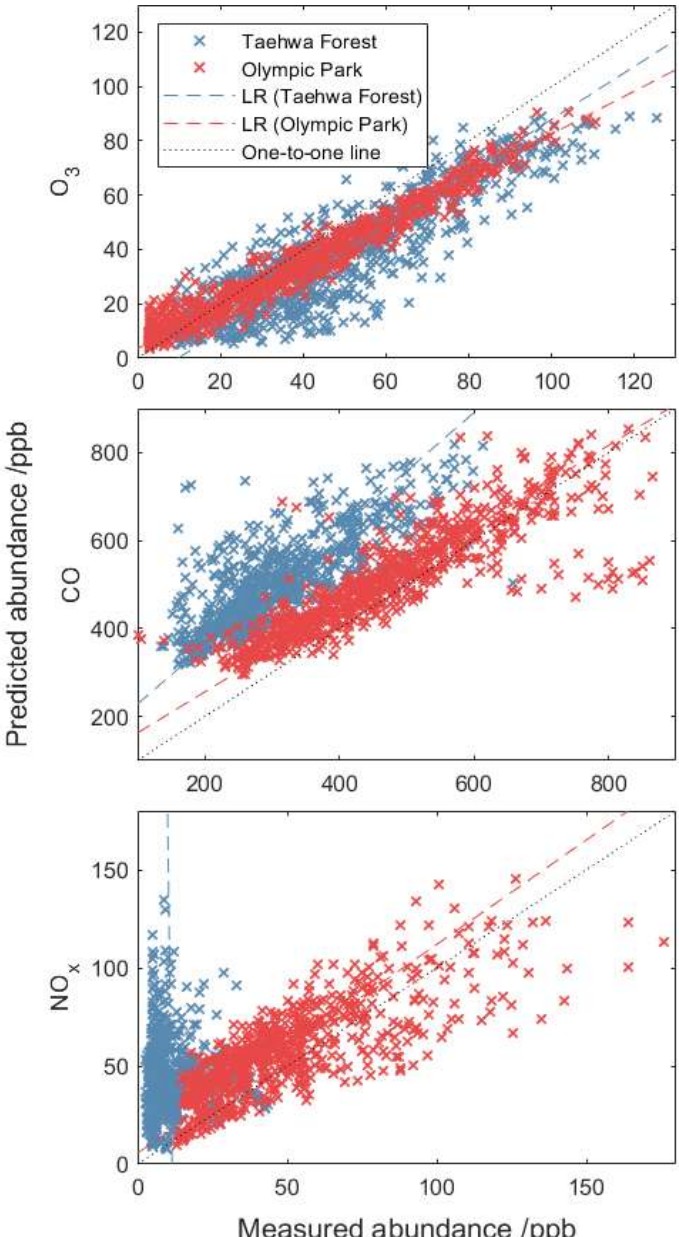


**Figure 5:** Predicted vs. measured abundances of the three key species at Olympic Park (red) and Taehwa forest (blue) research
stations. Predicted abundances are computed as point interpolations as per Equation (1). Dashed lines are linear regression
(LR) models fitted by ordinary least squares.

## 4.2 DC–8 comparison

Figure 6 (top panel) shows that the gridded surface–site predictions of the DC–8 $O_3$ observations are consistently lower than

observed but remain strongly correlated. CO predictions (Fig. 6, middle panel) show a consistent bias of around +100 ppb, but

otherwise capture the variability of the aircraft CO measurements reasonably well. $NO_x$ predictions (Fig. 6, bottom panel)
show a consistent positive bias along with randomness in the low measured range (<10 ppb). The gridded $O_3$ and CO
predictions are highly accurate ($R^2_{E2}$ = 80%) in grid cells with effective observation density ($Q$) exceeding ten, mainly sampled
in the Seoul Metropolitan Area (Fig. 1, right–hand side). These findings show that with enough ground information, our
gridded $O_3$ and CO datasets can predict upper PBL variability even in regions with intense small–scale emission heterogeneity.
$NO_x$ is exceptional, however, due to the rapid fall–off in abundance with altitude even within the PBL (Fig. A2 of Appendix
A, see also Fig. 2 from Kim et al., 2021). We believe that $O_3$ titration in the Seoul Metropolitan Area leads to a slight
underestimation in predicted variability as shown by a 10% increase in predictability using linear regression ($R^2_{E3}$ = 90%,
Table 1). We note however that the recurring flight patterns did not uniformly sample our grid domain and may have over– or
under–sampled some regions. Obtaining vertically averaged concentrations rather than surface values remains a challenge
given the substantial near–surface gradients inferred from Figs. 6 and A2, and suggests the need for vertically resolved
chemical and dynamical modelling.

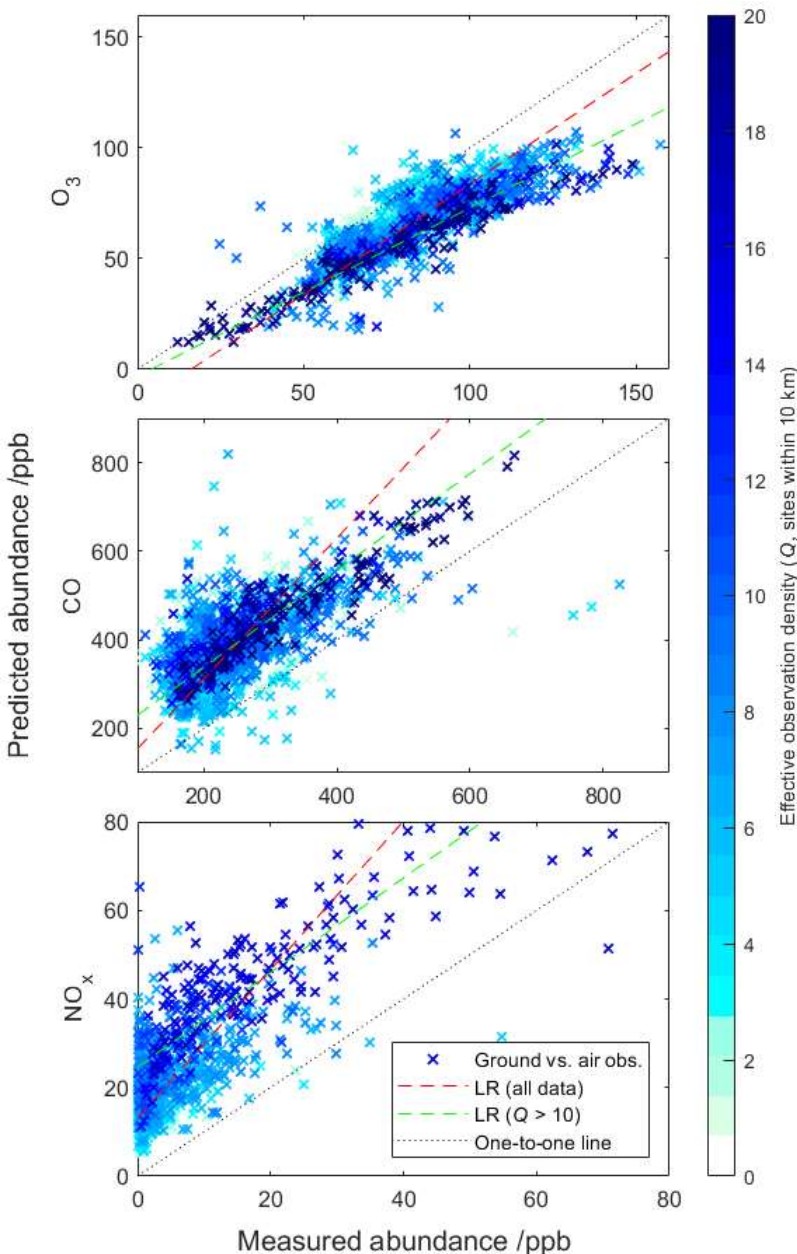


**Figure 6:** Comparison of 10 s DC–8 observations in the PBL and gridded (0.1°x0.1°) hourly ground station data. Each data
point represents the median of the contiguous aircraft transect through a grid cell (y–axis) and the median of the gridded ground
station data interpolated linearly in time to match the aircraft time of flight (x–axis).

## 5 Conclusions

We create gridded (0.1°x0.1°) observational datasets from NIER ground station measurements of air quality over South Korea. Unlike the arithmetic mean gridding technique, this method includes information from all nearby stations, including those outside the cell boundary, while mitigating sampling bias from site clustering. We identified significant mean absolute deviations between the IDW and arithmetic mean gridding techniques in *e.g.* the Seoul Metropolitan Area where IDW was most proved most accurate, prompting caution on the use of arithmetic mean gridding.

Our results suggest that the mean and variability of ground level $O_3$ was well captured over the whole of South Korea. For CO and $NO_x$, our LOOCV analysis revealed mean biases in certain NIER site predictions, but otherwise good prediction accuracy in most densely observed urban regions after the biases were subtracted. The well–predicted regions include the Seoul Metropolitan Area, Busan, Changwon, Daegu, and Cheongju, whereas prediction accuracy was poorer in the conjoined coastal cities of Gwangyang, Yeosu, and Suncheon, and in Ulsan; predictability in these regions would benefit from denser sampling. The aircraft comparisons confirm that the variability of $O_3$ and CO in the PBL are well captured from the surface stations; however, $NO_x$ vertical gradients in the PBL confound attempts to predict the aircraft $NO_x$ measurements.

Inverse Distance Weighting is susceptible to errors from over– or under–sampling of intense emission sources such as roadsides and industrial sectors. The characteristic concentrations of these regions may also be projected beyond the reach of the emission sources as seen in the high CO and $NO_x$ biases at Taehwa Forest. These error sources are not unique to IDW. Nevertheless, improvements in $NO_x$ predictability might be found with Land–Use regression and Machine Learning approaches as reviewed by Karroum *et al.* (2020), although these are outside the scope of the present paper. Such techniques can account for reactive $NO_x$ decay away from sources, potentially mitigating errors from source–sampling bias and over–projection. It would be interesting to compare the site predictabilities *vs.* sampling density for these alternate techniques, particularly in regions that were under–sampled according to LOOCV.

**6 Appendices**
Appendix A

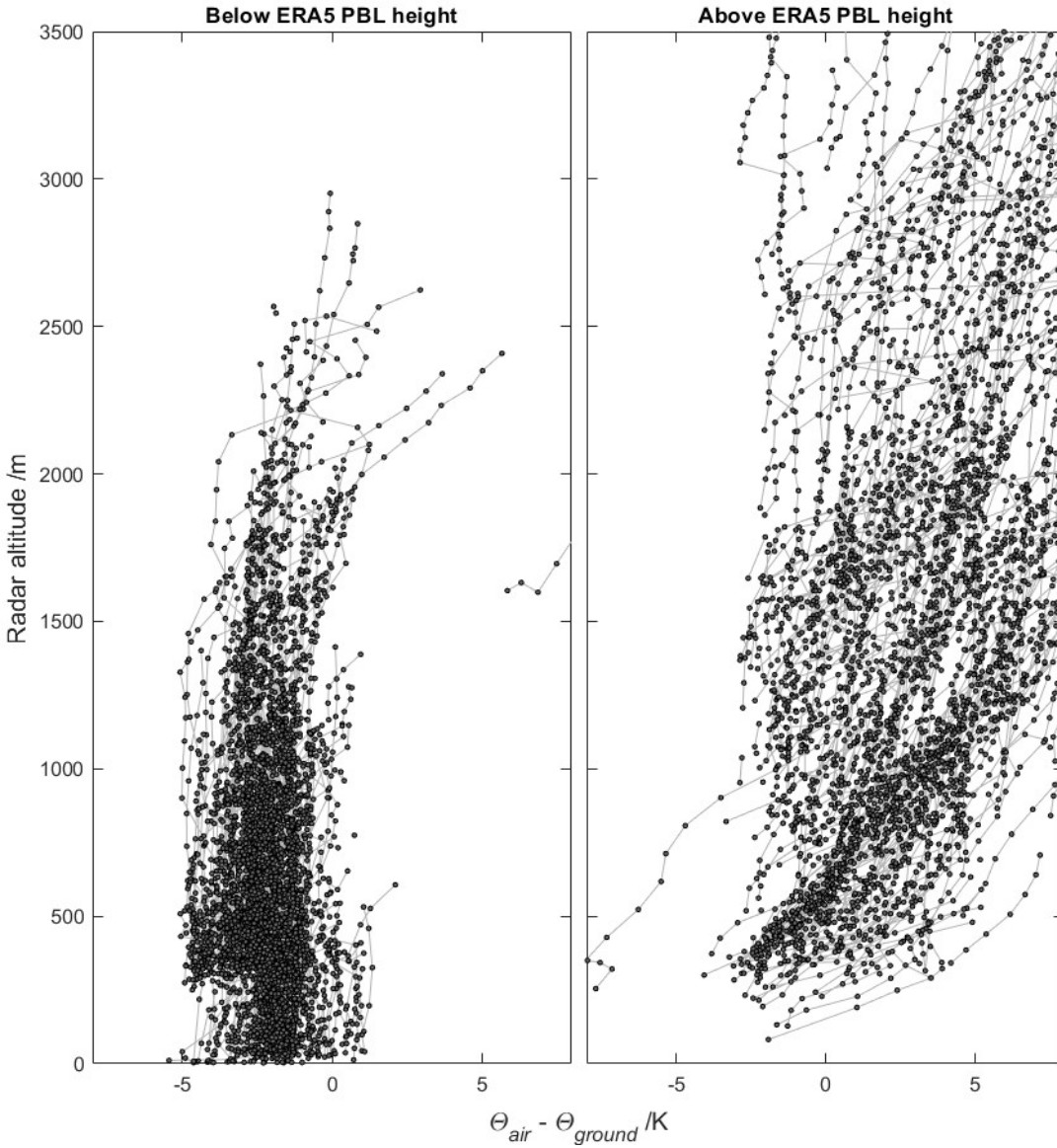


**Figure A1:** DC–8 10 s potential temperature ($\theta_{air}$) measurements (dots) in a half degree radius of Taehwa Forest research
station with gridded (0.25°x0.25°) surface potential temperature ($\theta_{ground}$) subtracted, taken below (**left**) and above (**right**) the
ERA5 designated PBL height. Lines connecting dots indicate contiguous transects, and all data was taken during ascent or
descent (aircraft vertical speed > 1 m s$^{-1}$). $\theta_{ground}$ was calculated using the ERA5 2 metre temperature and surface pressure
fields at native resolution (0.25°x0.25°, hourly), interpolated in time to match the aircraft time of flight.

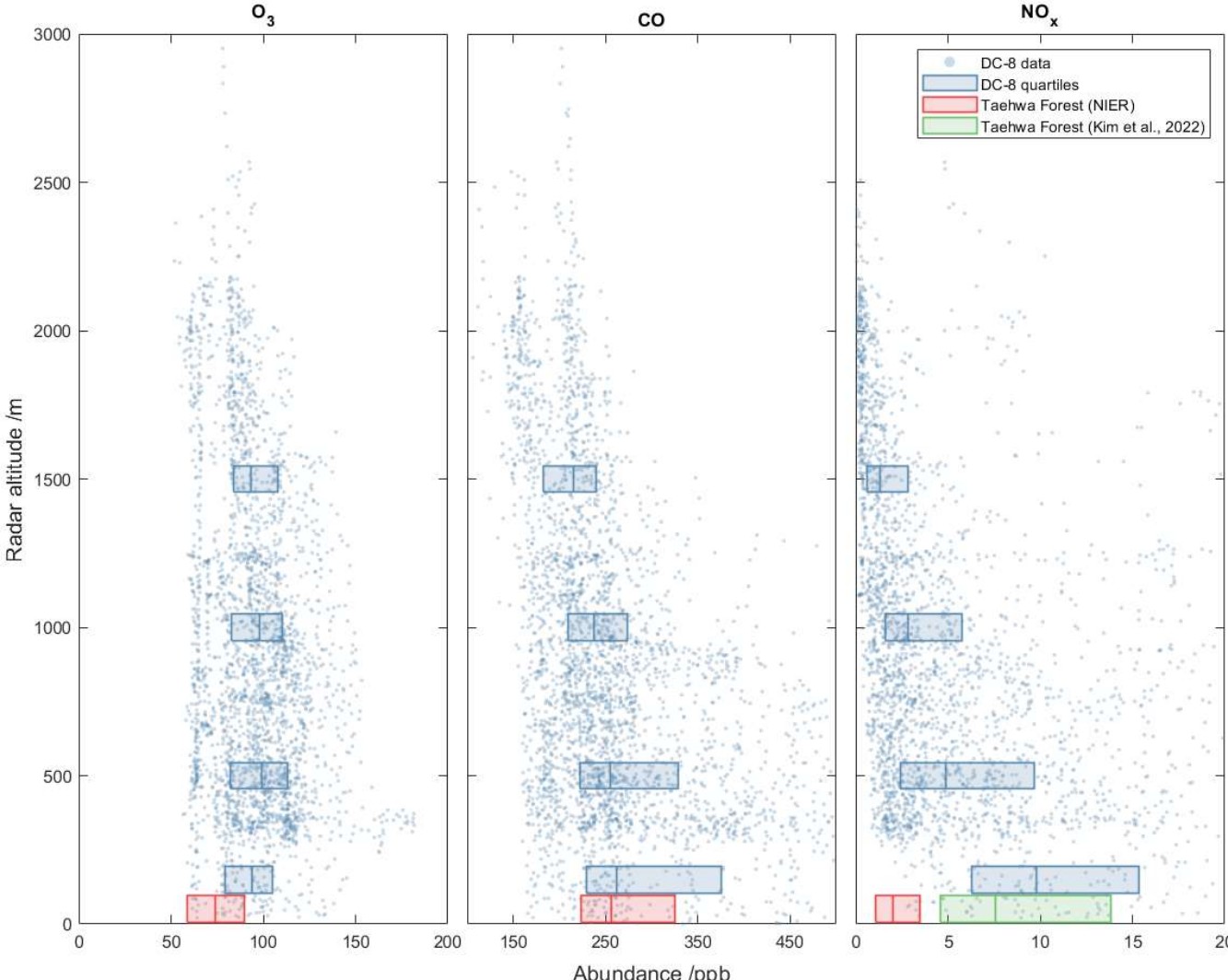

**Figure A2:** Vertical profiles of the DC–8 measured $O_3$ (**left**), CO (**middle**), and $NO_x$ (**right**) in the ERA5 PBL within a half degree radius of Taehwa Forest research station. All data is sampled between the hours of 12:00 and 17:00 LT, and quartiles are shown for aircraft data (blue) partitioned into altitude bins (0–250, 250–750, 750–1250, and 1250–1750 m) and for the available ground research station measurements at Taehwa Forest (red) supplemented by Kim et al., 2022 (green).

Appendix B

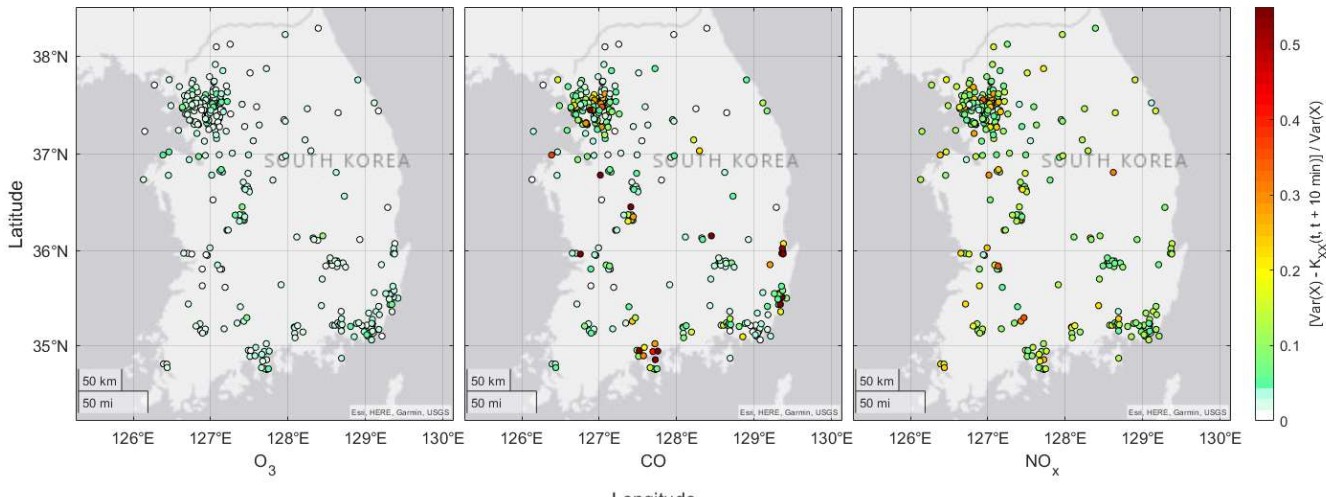

**Figure B:** The sub–ten–minute temporal variability of the quality–controlled $O_3$ (**left**), CO (**middle**), and $NO_x$ (**right**) site data
at five–minute resolution. Sub–ten minute variability for individual sites was calculated as the site autocovariance subject to a
ten–minute phase shift ($K_{XX}(t, t + 10\ min)$) subtracted from the total site variance ($Var(X)$), normalized by $Var(X)$.




















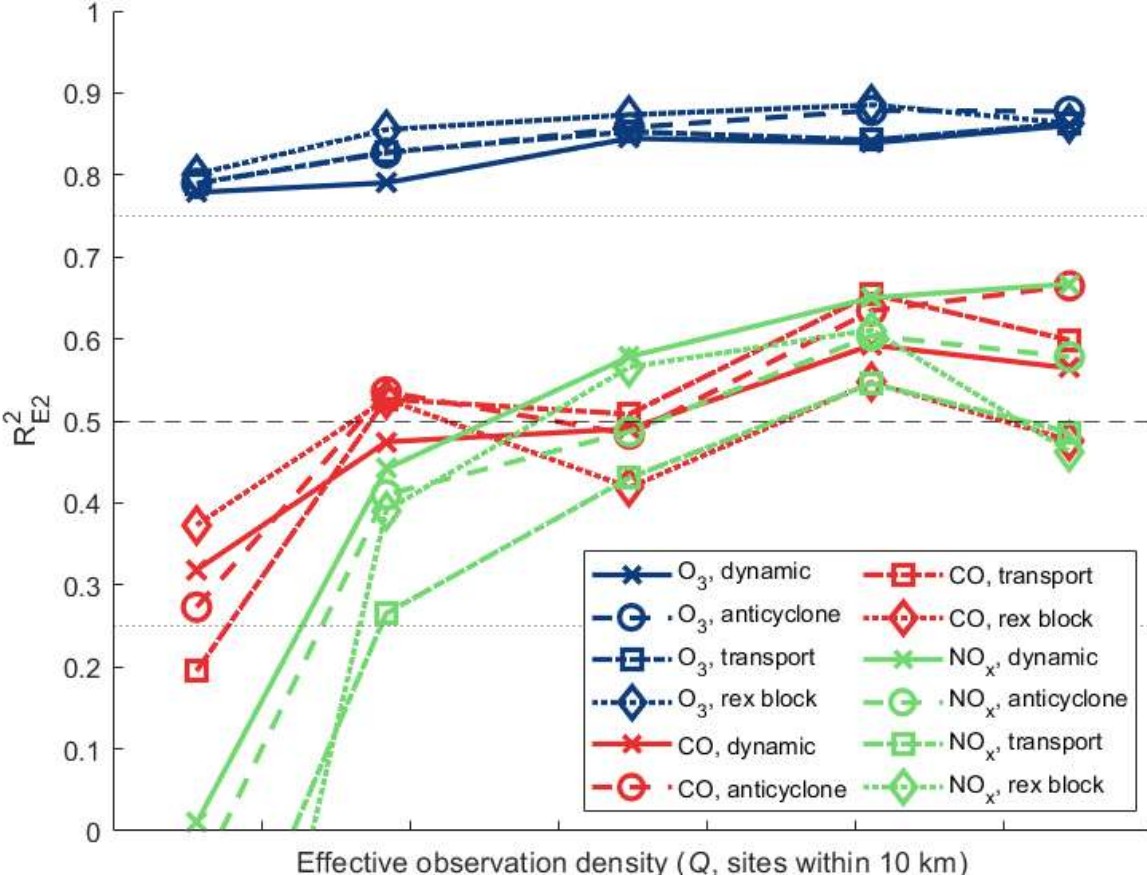

**Figure C1:** The median prediction accuracy ($R^2_{E2}$) of $O_3$ (**blue**), CO (**red**), and $NO_x$ (**green**) within percentile bins (0–20%,
20–40%, 40–60%, 60–80%, and 80–100%) during the meteorological phases (dynamic, anticyclone, transport, and rex
blocking) described by Peterson *et al*. (2021).

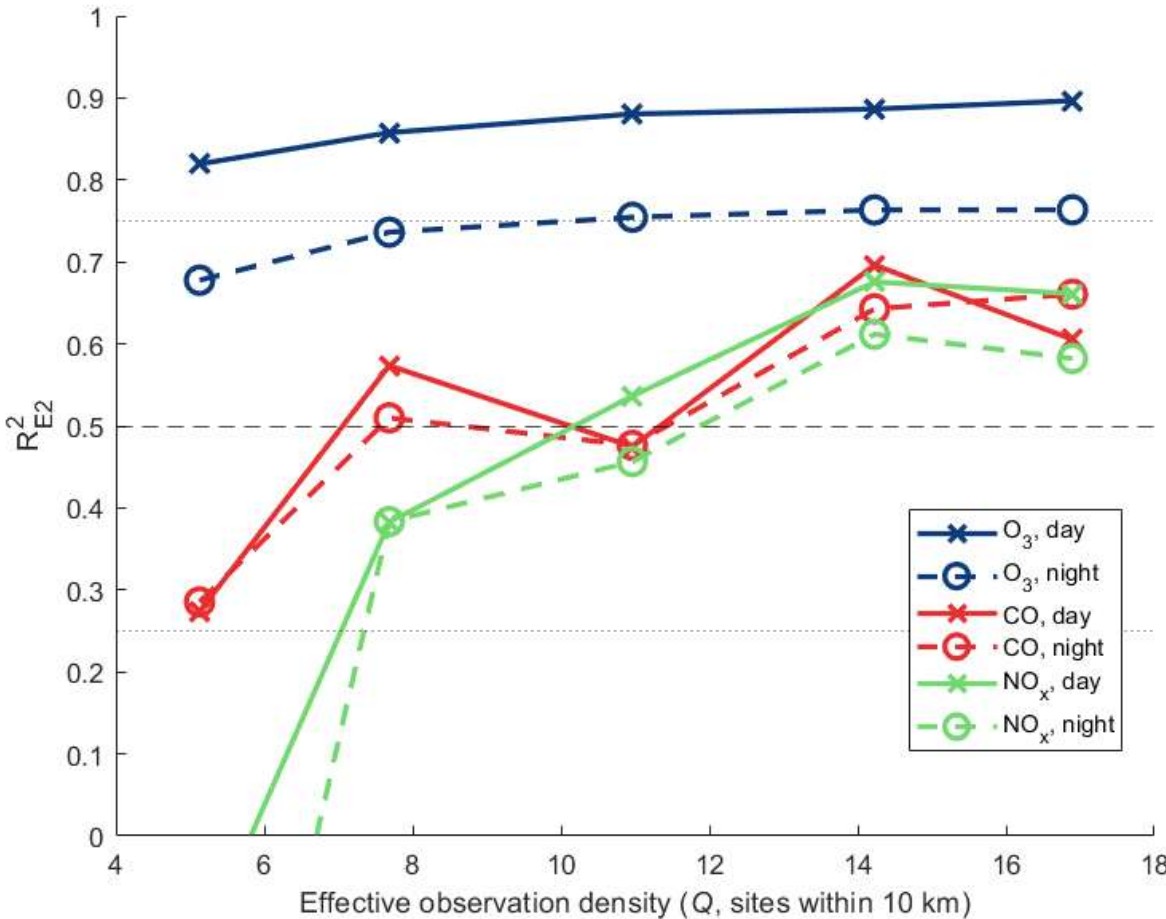

**Figure C2:** As in Fig. C1, but for daytime (07:00 to 20:00 LT incl.) and nighttime (21:00 to 06:00 LT incl.) data.
**Data Availability**
Gridded data products and the datasets used in analysis are available from Wilson, 2024:
https://doi.org/10.5061/dryad.sf7m0cgf5.
**Author contribution**
CW wrote the code to produce the datasets, codesigned and performed the analysis, drafted the manuscript, and responded to
the reviews. MP designed the methodology, codesigned the analysis, reviewed and edited the manuscript.
**Competing interests**
The authors declare that they have no conflict of interest.
**Acknowledgements**
This study was funded by NASA (# 80NSSC21K1454) and the National Science Foundation (NSF, # AGS-2135749). We
acknowledge NASA and NIER for providing the trace gas data used in this study and we are grateful to Kim Saewung for
guidance on the KORUS–AQ data usage.

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
