# Peer review of "Gridded surface O3, NOx, and CO abundances for model metrics from the South Korean ground station network"

_EGUsphere, 2024_

## Author Response (AR1)

**Review 1**

**We thank the reviewer once again for investing their time into our manuscript, and we detail how we addressed the comments in the upcoming manuscript version.**

*c1) This manuscript presented an IDW-based spatial interpolation method and an hourly gridded (0.1 x 0.1 deg) dataset for O3, CO, NOx. The gridded dataset was derived from the interpolated ground site observations in South Korea during the period of KORUS-AQ field campaign. The authors used this approach to mitigate the bias due to uneven density of the ground sites. The interpolation method and the gridded dataset were rigorously tested and analyzed in terms of bias and variability. The gridded dataset described in the manuscript will be useful to assess and improve models. The IDW-based interpolation approach is relatively straightforward and can also be used by researchers to better use the ground network observations. At the same time, it should be recognized that the IDW-based approach may not fully address the effect of microscale meteorological and local emissions, which both can be important under certain conditions. This reviewer believes it would benefit the readers if the authors can add more detailed discussions on the advantages and limitations the IDW-based approach, specifically discussing the statistical test results in the context of microscale meteorological conditions (e.g., wind speed and direction) and local emissions. Another important issue is the need to highlight the difference between weighted average and arithmetic average approaches in three grid cases with low, mid, and high Q values. This can be done by contrasting O3, CO, and NOx values between the gridded data presented in this manuscript and those computed from simple averages.*

**r1) We have added a discussion of the IDW vs. Arithmetic Mean techniques, noting significant differences in *e.g.* the Seoul Metropolitan Area, where IDW was previously shown to achieve good predictability.**

*Line 125: Compared with the arithmetic mean gridding, the gridded IDW observations show no significant mean bias. However, significant mean absolute deviations can be seen in the densely observed (Q > 10) Seoul Metropolitan Area and Southeast coastal cities. In such regions, the mean absolute deviations for $O_3$, CO, and $NO_x$ are 5 ppb, 58 ppb, and 9 ppb respectively. Conversely, the most sparsely measured regions show the best agreement between the two methods due to mutual sampling of a single station. Both methods are fundamentally limited by sampling density, particularly in urban centres with high spatial emission variability. The differences seen in the most densely observed regions highlight the instability of the arithmetic mean method with respect to grid size and boundary manipulation.*

*c2) Line 90 – 91: The authors should clarify how the correlation was computed between different sites and discuss if different sites have similar temporal variation patterns with a phase shift.*

**r2) We have clarified how the correlation was computed and added further analysis of the site autocorrelations.**

*Added Appendix B: Figure B*

*Line 94: In our work we find minimal Pearson's correlation between…*

*Line 97: We examine the autocovariance functions of the sites to characterize the temporal variability of each key species at five–minute resolution. The autocovariance function for a given phase shift (e.g. ten minutes) is defined as the covariance of a time series of data with itself, but phase–shifted by ten minutes. The $O_3$ site autocovariance functions feature strong diurnal cycles, preserving on average half of the respective site variances for a twenty–four–hour phase shift, compared with 35% for CO and $NO_x$. Sub–ten–minute variability accounts for 3% of the $O_3$ site variance and 12% of the CO and $NO_x$ variance on average. Heterogeneous emissions and micrometeorology may be attributed to some of the CO and $NO_x$ short timescale variability seen in some Seoul sites and many CO–measuring sites in Gwangyang–Yeosu–Suncheon zone, while rural sites suggest a possible instrumental noise component to the $NO_x$ variability (See Fig. B of Appendix B).*

*c3) Section 3.2: This reviewer would like to raise a question if the better results obtained for $O_3$ is partially attributed to that $O_3$ abundance is not directly influenced by local emissions while the CO and $NO_x$ at a given site can be substantially affected by local emissions. It is possible that certain emission events are seen only in a few sites and the IDW interpolation would not be able to predict these observations in the leave-one-out tests. In this context, it would be helpful if the authors can state the limitation of the IDW interpolation under certain conditions.*

**r3) We agree with the reviewer in our arguments on lines 164 to 165, but have added the clarification that $O_3$ is not directly emitted, unlike the other species. We added a description of the IDW limitations in the conclusion.**

*Line 84: …implying they are driven by very small scale (<1 km) local emissions in contrast to $O_3$, which is not emitted directly.*

*Figure A3 promoted to Figure 4 and updated to show the difference between LOOCV predictability and predictability using all stations.*

*Line 188: When we compare the LOOCV performance of NOx with the complete interpolation (i.e., no stations omitted, see bottom panel of Fig. 4), we see R2E2 scores change by > 0.4 at e.g. rural sites, the Busan shoreline, and the manufacturing district sites of Northern Yeosu and Eastern Gwangyang. These discrepancies indicate under–sampling of NOx across rural South Korea and in some urban districts with locally contrasting emission activity.*

**c4) Section 4.2: It should be stated in this section that the DC-8 sampling may not be representative of the grids due to limitation of the flight patterns.**

**r4) We updated our manuscript to acknowledge this fact.**

*Line 267: We note however that the recurring flight patterns did not uniformly sample our grid domain and may have over– or under–sampled some regions.*

**c5) The conclusion section should highlight the advantages and limitations of the IDW interpolation approach.**

**r5) We have added a discussion of IDW vs. arithmetic mean technique in the conclusion and described the limitations of IDW, noting how some alternative techniques could address these limitations.**

*Line 277: Unlike the arithmetic mean gridding technique, this method includes information from all nearby stations, including those outside the cell boundary, while mitigating sampling bias from site clustering. We identified significant mean absolute deviations between the IDW and arithmetic mean gridding techniques in e.g. the Seoul Metropolitan Area where IDW was most proved most accurate, prompting caution on the use of arithmetic mean gridding.*

*Line 288: Inverse Distance Weighting is susceptible to errors from over– or under–sampling of intense emission sources such as roadsides and industrial sectors. The characteristic concentrations of these regions may also be projected beyond the reach of the emission sources as seen in the high CO and NOx biases at Taehwa Forest. These error sources are not unique to IDW. Nevertheless, improvements in NOx predictability might be found with Land–Use regression and Machine Learning approaches as reviewed by Karroum et al. (2020), although these are outside the scope of the present paper. Such techniques can account for reactive NOx decay away from sources, potentially mitigating errors from source–sampling bias and over–projection. It would be interesting to compare the site predictabilities vs. sampling density for these alternate techniques, particularly in regions that were under–sampled according to LOOCV.*

*c6) The authors should consider adding global attributes and variable attributes (e.g., units) to the gridded netCDF file and make the file more CF compliant, e.g., using CF variable names, like time, lat, and lon. This will enhance the (re)usability and interoperability of the hourly gridded dataset.*

r6) Great idea, thanks. We have updated our datasets to better comply with the CF standard. We use more conventional variable and dimension aliases (e.g. *lat*, *lon*, *time*) along with *units*, *long_name*, and *description* attributes. We have added global attributes that specify where found the data and how we processed it.

**Review 2**

*Traditionally, simple grid-cell averages have been used to test and analyze the results of regional air quality models. However, spatial heterogeneity, local isolated emission sources and data uncertainties have often raised questions about the representativeness of this approach. This study represents a valuable and scientifically innovative effort to assess the validity of such methods and to explore potential alternatives. The results are not only in line with the aims of the journal, but also have significant potential for wider scientific impact, as they can contribute to critical assessments such as determining the suitability of measurement sites, identifying observational biases, and estimating contributions from local isolated sources. Nevertheless, this manuscript raises several important issues that require further clarification and resolution before publication.*

We thank you for your insights and your recognition of the value in gridding the site data. We offer our responses below.

*c1) Chemical species such as O3, CO and NOx are known to exhibit significant spatial variability due to various chemical and physical factors, as well as local emission sources and observational biases. To better highlight the performance of the method proposed by the authors for these chemical species, would it not be more effective to compare them to meteorological or physical variables such as temperature, which tend to have less spatial variability and are less prone to observational bias? Analyzing these stable variables using the same method and using them as a reference might provide a clearer context for interpreting the variability of the chemical species, more in line with the authors' aims.*

r1) Use of meteorological quantities is interesting but these are not available for the sites, and it brings new uncertainty as to the cause of biases that we cannot ascertain (e.g., surface and building albedo, urban heat island effects, orography). In a new

**Appendicised Figure C2 added to the paper, we examine the site predictability during daytime vs. nighttime. Daytime O3 predictability is higher than nighttime across South Korea by at least +10% points , and similarly a +5% increase is seen in urban (high emission) areas for NOx. This new work suggests that the comparatively turbulent daytime PBL mixes local (NOx) emissions more efficiently while nighttime stability produces sharper (and less predictable) concentration gradients. Discussion has been added to Section 3.2.1.**

*Appendix C, Figure C1 and C2 added to the manuscript.*

*Line 209: An alternative test of meteorological influence is daytime (07:00 to 20:00 LT) vs. nighttime (21:00 to 06:00 LT) predictability, i.e., prevailingly turbulent vs. stably stratified atmospheric surface conditions. Ozone and NOx showed greater LOOCV predictability (R2E2) during daytime than nighttime by around +0.1 and +0.05 respectively, but not CO (see Fig. C2). More efficient turbulent mixing during daytime vs. nighttime likely smoothed some of the small–scale emission heterogeneity, resulting in more predictable fields.*

*c2) In many cases, treating O3 and NO2 together as Ox (= O3 + NO2) rather than separately provides more robust results when comparing model outputs with observations. Would it not be valuable to assess how the results differ when these species are analyzed as a single group (Ox) compared to analyzing O3 and NOx separately? Such an evaluation could provide additional insight into the robustness and reliability of the proposed method.*

**r2) We kept O3 and NOx separate because they are fundamentally separate chemical-model diagnostics and their spatial variability has different patterns. Further, Ox is a derived quantity for the models and sometimes even includes higher nitrates (e.g., NO3 = 2 Ox).**

*c3) I agree that the method proposed in this study produces significantly better results compared to simple grid-cell averages. However, the conclusions drawn regarding the predictive accuracy of IDW interpolation and the effectiveness of leave-one-out cross-validation (LOOCV) raise some questions. While LOOCV provides a useful validation of the interpolation method, does it sufficiently address the inherent limitations of IDW in capturing spatial heterogeneity and clustering effects in areas of uneven data distribution and local outliers? For example, poor prediction accuracy in specific regions such as Gwangyang, Yeosu, Suncheon, and Ulsan may indicate limitations of the IDW approach itself, beyond what bias corrections or LOOCV validation can mitigate for isolated outliers. Would alternative interpolation methods, such as kriging or hybrid approaches, incorporating geostatistical models with spatial emission*

*source distributions, better address these challenges?  Such a comparison would help determine whether IDW is indeed the most suitable choice for this dataset*

**r3) A full parallel analysis with kriging and hybrid methods is beyond the scope of the work here, although it would be informative. It is reasonably certain that IDW (and other methods) outperform the arithmetic mean gridding approach that much of the community has been using. We explained why Kriging methods are inappropriate, essentially because we can't establish a relationship between covariance and proximity for any species. It would be valuable to apply a land-use regression model for NOx in particular, since this could fix the projection of roadside NOx levels into wilderness sites such as Taehwa Forest and establish 1) whether the interpolation improves in coastal cities, and 2) how this improvement changes the gridded averages. Such endeavours require data and resources that are not immediately available to us.**

*See Line 288, r5) of Review 1.*

*c4) Page 2 line 37-38: It is unclear from the manuscript whether the data from the AirKorea monitoring network were obtained directly from the official data center such as NIER, where they are subject to QA/QC management, or whether they were downloaded directly from the AirKorea website (https://www.airkorea.or.kr/eng). If the data were obtained from the AirKorea website, it is important to note that the data available there are real-time observations with minimal QA/QC and may contain some errors. This could potentially contribute to the observed elevated $E1(t)$ values. Clarification of this aspect seems necessary.*

**r4) This is an important distinction. https://www.airkorea.or.kr/ has often been cited as the original source of NIER air quality data (e.g. Crawford *et al*., 2021), but we could not access the data directly from the website, and instead used the data from the KORUS-AQ period stored in the NASA archive: https://asdc.larc.nasa.gov/project/KORUS-AQ. For convenience, we have now published the raw (5 minute) data along with our own QA/QC flags in our repository ((https://doi.org/10.5061/dryad.sf7m0cgf5) and indicated this in Section 2.1.**

*Line 38: All KORUS–AQ datasets introduced in this section including the raw five–minute NIER station data are available via https://doi.org/10.5067/Suborbital/KORUSAQ/DATA01. The NIER station data is access–controlled, but we record our processed hourly–averaged NIER station data and our quality control flags for the raw data in our repository (see Wilson, 2024).*

*It would be beneficial if the methods and results sections provided more detailed explanations on the following points:*

*c5) A comparison of prediction errors across different regions, particularly between data-dense and data-sparse areas, to evaluate the robustness of the interpolation method under varying observation densities. Perhaps Taehwa and Olympic Park can be used for this purpose?*

**r5) Our new additions to Figure A3 provide interesting insights into the interpolation robustness. By examining the improvement of the complete interpolation over the LOOCV interpolation, we highlight regions that were sufficiently sampled (little improvement, *e.g.* the Seoul Metropolitan Area) *vs*. undersampled (*e.g.* rural sites and some city districts). This discussion has been included in Section 3.2.1.**

*Figure A3 promoted to Figure 4 and extra panels added; see r3) of review 1.*

*c6) A clear explanation of how missing or erroneous observations were handled during the analysis. Additionally, an assessment of whether the chosen approach introduces bias into the interpolation results would strengthen the reliability of the conclusions.*

**r6) This was an oversight in our methodology discussion: we were not explicit in how missing or erroneous data were treated by the interpolation and subsequent analysis. We have clarified this in Section 2.1. In summary, stations with missing data (or errors that we flagged) were excluded from the interpolation on an hourly basis and discounted from the cluster radius (D) of other stations for declustering purposes. Comparing LOOCV predictions with the complete interpolation provides a metric on the impact of station malfunction on predictability, which we have now charted in Figure A3.**

*Line 45: We calculate hourly median readings centred on the hour for each station, but omit clearly erroneous O3 and NOx dropouts from the average.*

*Line 107: The greater density of observations in some regions creates a source of oversampling bias. Schnell et al. (2014) address this clustering effect by reducing all station weights by $M\_k$, the number of other stations within distance D of site k, discounting sites with missing readings on an hourly basis.*

*See r3) of Review 1 for discussion on insight into biased regions (Figure 4).*

*c7) Sensitivity analyses on the parameters (β, D, L) to ensure that the selected values are truly optimal and not overly reliant on specific conditions within the dataset.*

**r7) We have added a statement on how we optimised the parameters: a recursive approach where we fix D and iterate through β, then fix β and iterate D, until a 2D minimum is reached (L is a time-saving constraint that does not significantly affect the global interpolation error above a certain threshold). We omitted sensitivity analysis because the 2D minimae are very similar for all species and shallow, i.e., the global interpolation error is not sensitive to changing the parameters.**

*Line 118: We optimize β and D for each key species Z by randomly removing a fifth of the stations from the algorithm and then predicting the abundance at each missing station k', adjusting each parameter until a 2D minimum was reached.*

*c8) page 7. Line 168 Jiju -> Jinju ?*

**r8) Thank you for spotting this, it has been corrected.**

*Line 188: …and in the southern coastal cities of Gwangyang, Yeosu, Suncheon, Jinju, and Ulsan (no NOx data).*